# Associations between Maternal Cadmium Exposure with Risk of Preterm Birth and Low after Birth Weight Effect of Mediterranean Diet Adherence on Affected Prenatal Outcomes

**DOI:** 10.3390/toxics8040090

**Published:** 2020-10-20

**Authors:** Sarah Gonzalez-Nahm, Kiran Nihlani, John S. House, Rachel L. Maguire, Harlyn G. Skinner, Cathrine Hoyo

**Affiliations:** 1Department of Nutrition, University of Massachusetts Amherst, Amherst, MA 01003, USA; 2Department of Statistics, University of Pittsburgh, Pittsburgh, PA 15260, USA; kiran.nihlani@pitt.edu; 3National Institute of Environmental Health Sciences, Durham, NC 27709, USA; john.house@nih.gov; 4Department of Biological Sciences, North Carolina State University, Raleigh, NC 27606, USA; rlmaguir@ncsu.edu (R.L.M.); harlyn.skinner@ncsu.edu (H.G.S.); choyo@ncsu.edu (C.H.)

**Keywords:** cadmium, heavy metals, birth weight, preterm birth, diet pattern, Mediterranean diet, pregnancy

## Abstract

Prenatal cadmium exposure at non-occupational levels has been associated with poor birth outcomes. The intake of essential metals, such as iron and selenium, may mitigate cadmium exposure effects. However, at high levels, these metals can be toxic. The role of dietary patterns rich in these metals is less studied. We used a linear and logistic regression in a cohort of 185 mother–infant pairs to assess if a Mediterranean diet pattern during pregnancy modified the associations between prenatal cadmium exposure and (1) birth weight and (2) preterm birth. We found that increased cadmium exposure during pregnancy was associated with lower birth weight (β = −210.4; 95% CI: −332.0, −88.8; *p* = 0.008) and preterm birth (OR = 0.11; 95% CI: 0.01, 0.72; *p* = 0.04); however, these associations were comparable in offspring born to women reporting high adherence to a Mediterranean diet (β = −274.95; 95% CI: −701.17, 151.26; *p* = 0.20) and those with low adherence (β = −64.76; 95% CI: −359.90, 230.37; *p* = 0.66). While the small sample size limits inference, our findings suggest that adherence to a Mediterranean dietary pattern may not mitigate cadmium exposure effects. Given the multiple organs targeted by cadmium and its slow excretion rate, larger studies are required to clarify these findings.

## 1. Introduction

Although cadmium is a naturally occurring heavy metal, its increased use in numerous industrial applications has made it one of the most abundant environmental pollutants present in atmospheric, terrestrial, and aquatic systems [1,2]. Cadmium is classified as a probable carcinogen [3] and its lack of degradation in the environment facilitates its persistence and enables sustained human exposure [2,4]. While contaminated air from industrial processes is the most cited source of occupational exposure [2], non-occupational cadmium exposure can occur through the inhalation of tobacco smoke and dust. Cadmium is also present in some commercial fertilizers [5,6,7,8] and contamination of agricultural soils results in the ingestion of cadmium through dietary staples [2,4]. In the US and other regulated societies, dietary cadmium intake is estimated at ~1 µg/day [1,9]. The slow excretion of cadmium leads to accumulation in the body over time [4].

Early-life cadmium exposure in children and pregnant women has been associated with low birth weight (either due to growth restriction or shorter gestation) [10,11,12,13], childhood disorders including neurodevelopmental disorders, and indicators of metabolic dysfunction such as obesity—which have been recapitulated in zebrafish [14]. Low birth weight and preterm birth are public health concerns, as they are risk factors for early mortality and the onset of later disease and co-morbidities [15]. Approximately 8% of infants in the US are born with a low birth weight (≤2500 g) [16] and approximately 10% of infants are born preterm (prior to 37 weeks gestation) [17] with increased frequencies for African-American infants [17] The management of low birth weight and preterm birth and associated co-morbidities poses a large financial burden on families and the healthcare system [18].

Dietary supplementation with essential metals including iron, calcium and selenium, has been recommended by multiple environmental health agencies to mitigate the effects of cadmium exposure, in part because cadmium influx occurs with metal transporters for these essential metals [19,20,21]. However, appropriate doses are unclear, as these essential metals can be toxic at high doses [20,21]. Polyphenols and other antioxidants in the diet have also shown the potential to reduce the negative consequences of cadmium exposure [20,21]. As humans typically consume combinations of nutrients as a part of meals or whole foods, the study of dietary patterns is an important tool to understand how public health recommendations can help reduce the risk from prenatal cadmium exposure. The Mediterranean diet pattern which is characterized by a high intake of iron, selenium, and antioxidants can be easily studied and translated to public health guidelines. Maternal adherence to a Mediterranean diet during pregnancy has been found to be associated with a reduced risk of gestational diabetes [22], normal birth weight [23,24,25,26] and longer gestational age [26,27], as well as favorable behavioral patterns [28], and other positive child outcomes [26]. Although there is an abundance of evidence suggesting multiple health benefits from adhering to a Mediterranean diet pattern [29,30,31,32,33], the potential role of this diet in mitigating the effects of prenatal cadmium exposure and birth outcomes has not yet been described. These analyses aim to explore the effect measure modification of Mediterranean diet pattern adherence during pregnancy in the association between elevated prenatal cadmium exposure and birth outcomes, including birth weight and preterm birth, and exploratory analyses of the association between prenatal cadmium exposure and Apgar scores and infant ponderal index at birth.

## 2. Materials and Methods

We used data from participants of the Newborn Epigenetics Study (NEST), a cohort of women–infant dyads from central North Carolina. Enrollment details have been described elsewhere [34]. In brief, 1700 women enrolled during pregnancy between 2009 and 2011 at qualifying prenatal clinics. Women met the following inclusion criteria: 18 years of age or older, plan to deliver in one of two birthing facilities in Durham county, and English or Spanish speaking. We excluded women who planned to give up custody of their child and those who did not carry offspring to term. Of the 1700 enrolled, 1304 remained after additional exclusions (*n* = 115 experienced a fetal death, *n* = 281 refused further participation or an inability to follow-up with the participant). We collected blood and obtained cadmium measures for the first *n* = 310 women. Of the 310, *n* = 298 had non-missing values for birth weight, and *n* = 185 women completed a food frequency questionnaire (FFQ). The median gestational age at enrollment was 11–12 weeks. We have previously shown that the 310 mother–infant pairs in whom cadmium was measured did not vary significantly from the remainder of the cohort [14]. The *n* = 185 on whom FFQ data were available also did not differ from the 310 with respect to sex, race/ethnicity and maternal obesity distribution (*p* > 0.05). Those included were, however, more likely to have a higher educational level and older maternal age at delivery (*p* < 0.05). These factors were adjusted for in the analysis. The women in our cohort were not significantly different with respect to covariates from the women in the overall NEST cohort (*p* > 0.05). This study was approved by the Duke University Institutional Review Board (#Pro00014548) on 19 February 2020.

### 2.1. Cadmium Exposure

We measured cadmium in whole blood for the first 310 enrolled women at a median gestation age of 12 weeks, using ICP-MS and methods previously described in detail [11,13]. Because cadmium co-occurs with other environmental pollutants [35], we also measured lead and arsenic. Briefly, we measured prenatal cadmium concentrations in whole blood donated at enrollment as nanograms per gram (ng/g; 1000 ng/g = 1035 ng/μL) of blood weight using well-accepted solution-based ICP-MS methods. We homogenized temperature equilibrated whole blood samples (0.2 mL) and pipetted them into a trace metal-clean test tube. We used a calibrated mass balance to confirm the samples gravimetrically to ±0.001mg, and we spiked samples with internal standards consisting of known quantities (10 and 1 ng/g, respectively) of indium (In) and bismuth (Bi) (SCP Science, USA), used to correct for instrument drift. We then diluted the solutions with water purified to 18.2 MΩ/cm resistance, which we will refer to as Milli-Q water (Millipore, Bedford, MA, USA) and acidified the solutions using ultra-pure 12.4 mol/L hydrochloric acid to result in a final concentration of 2% hydrochloric acid (by volume). We prepared all standards, including aliquots of the certified NIST 955c, and procedural blanks using the same process. We measured Cd concentrations using a Perkin Elmer DRC II (Dynamic Reaction Cell) axial field ICP-MS at Duke School of the environment, Durham, NC, USA. Calibration standards used to assess metals in blood included aliquots of Milli-Q water, and NIST 955c SRM spiked with known quantities of each metal in a linear range from 0.025 to 10 ng/g. We prepared standards from 1000 mg/L single element standards (SCP Science, USA). We calculated method detection limits (MDLs) consistent with the two-step approach using the t99SLLMV method (USEPA, 1993) at 99% CI (*t* = 3.71). The MDLs generated values of 0.006, 0.005, and 0.071 μg/dL, for cadmium, lead and arsenic, respectively. The thresholds of detection (LODs) were 0.002, 0.002, and 0.022 μg/dL, for Cd, Pb and As, respectively, and limits of quantification (LOQs) (according to Long and Winefordner, 1983) were 0.0007, 0.0006, and 0.0073 μg/dL for Cd, Pb, and As, respectively. The number of samples below the LOD for Cd, Pb, and As were two, two, and one, respectively.

### 2.2. Mediterranean Diet

We measured overall diet using a modified food frequency questionnaire (FFQ) [36] at enrollment. Women were asked to report their usual intake over the past 3 months, allowing us to capture the periconceptional period. We scored women’s diets using the data-driven Mediterranean Diet Score (MDS) [37]. The MDS assesses adherence to a Mediterranean diet pattern based on the reported intake of foods that are deemed to be beneficial: fruit, vegetables, fish, dairy, whole grains, legumes, nuts, and monounsaturated fatty acids, and foods that are deemed detrimental: meat. We excluded alcohol from the diet score, as alcohol is not generally recommended during pregnancy and the reported alcohol intake in our cohort was low. Women who reported an intake of a beneficial foods at or above the median for the study population received a score of 1 and 0 otherwise. Those who reported an intake of detrimental foods below the median received a score of 1 or 0 otherwise. The MDS ranges from 0 to 9, with 0 representing the lowest possible adherence to a Mediterranean diet pattern and 9 representing the highest adherence to a Mediterranean diet pattern. We assessed maternal Mediterranean diet adherence as low (MDS at or below 4) and high (MDS above 4).

### 2.3. Birth Outcomes

At delivery, we abstracted parturition data, including infant birth weight and gestational age, from medical records. We used standard definitions for low birth weight (≤2500 g) and preterm birth (<37 weeks). We assessed birth weight continuously and preterm birth categorically (>37 weeks gestation/≤37 weeks gestation). We assessed infant Apgar scores continuously (1–10), with a higher score reflecting a greater level of health at birth. We also assessed birth length (cm), and derived the infant ponderal index (PI) at birth. PI is a measure of the proportionality of body growth and is calculated using the formula: weight (g) × (100/length (cm^3^)).

### 2.4. Statistical Analysis

We used a linear regression to assess the association between elevated cadmium exposure during pregnancy and birth weight, and a logistic regression to assess the association between elevated cadmium exposure and preterm birth. In our study, cadmium was severely right-skewed. Despite log transformation, the skewness did not improve (results using the continuous log Cd variables are available in Appendix A). As cadmium exposure is ubiquitous, we assessed cadmium in quartiles and defined high cadmium exposure as having a cadmium blood level in the highest quartile (mean Cd (ng/g) per quartile: 25th percentile: 0.12, 50th percentile: 0.24, 75th percentile: 0.46). We identified potential confounders a priori based on the literature and substantive knowledge and selected a final set of confounders using Bayesian Information Criteria (BIC). We included prepregnancy BMI, smoking during pregnancy, and sex of the infant as confounders in our final models. In our analysis of birth weight, we also included gestational age as a confounder and in our analysis of preterm birth we included birth weight as a confounder in the model. We assessed effect measure modification by Mediterranean diet adherence by including an interaction term in our models and by stratification of high and low maternal Mediterranean diet adherence. Effect measure modification analyses are limited to the 185 women who completed first trimester FFQs and had cadmium measures available. We conducted Appendix A to explore possible changes in the association between prenatal cadmium exposure and birth outcomes using different cut points for high and low Mediterranean diet adherence. Additionally, we explored the association between prenatal cadmium exposure and (1) infant Apgar score, (2) infant ponderal index at birth.

## 3. Results

We present study participant demographic factors overall, and by birth weight (Table 1) and Mediterranean diet adherence (Table 2). Of the 298 women included in our sample, 36% were Black, 28% were white, 32% were Latina or Hispanic, and 4% were of other race/ethnicities. Approximately one-quarter of women had a BMI of 30 or greater, and about 75% of obese women were either African American or Hispanic. Over half of our sample (55%) had a high school diploma. Among infants in our study, 6% were born prior to 37 weeks gestation and 5.7% weighed 2500 g or less at birth. Approximately 20% of mothers in our study reported smoking during pregnancy. The median (IQR) of cadmium concentration in blood was 0.24 (0.34) ug/g of blood weight, and MDS scores were normally distributed and ranged from 0 to 9 and approximately half of the women in our study had a score at or below 4. Women who did not deliver a low birth weight infant were more likely to be white, non-smokers, and have a college degree.

### 3.1. Cadmium Exposure and Birth Weight

Results of the associations between cadmium and birth weight are summarized in Table 3. After adjustment for prepregnancy BMI, smoking during pregnancy, gestational age, and sex of the infant, we observed that the women in the highest quartile of cadmium exposure during the prenatal period had infants whose birth weights were 210 g lower (*β* = −210.4; 95% CI: −332.0, −88.8; *p* = 0.008) than those in the lower three quartiles. Further including other co-occurring metals, lead and arsenic, that also have been linked to lower birth weights did not alter these associations. We explored removing gestational age from these models, as it may be on the causal pathway. This somewhat attenuated the association, however, it remained statistically significant (*β* = −161.6; 95% CI: −311.3, −11.9; *p* = 0.04). Further adjusting the co-occurrence of other metals, including lead or arsenic, did not materially alter these findings (data not shown).

We also regressed gestational age at birth on cadmium exposure, controlling for the same covariates (Table 4). We found that preterm birth is marginally associated with prenatal cadmium exposure (*β* = −0.11; 95% CI: 0.01, 0.72; *p* = 0.04) (Table 2). We included birth weight as a confounder in our main analysis and explored the effect of removing it in a Appendix A. When birth weight was removed from the model, the association between elevated prenatal cadmium exposure and preterm birth was no longer statistically significant (*β* = 1.2; 95% CI: 0.37, 3.31; *p* = 0.74). Again, further adjustment for the co-occurring metals, lead and arsenic, did not alter these findings, suggesting that growth restriction, rather than preterm birth, may be the major contributor to these birth outcomes.

### 3.2. Stratification by Mediterranean Diet Adherence

To determine whether these associations were modified by adherence to a Mediterranean diet pattern, we first dichotomized the MDS below the median (a score of 4 of 9) among the *n* = 185 of 310 pregnant women who also completed the food frequency questionnaire. We examined cadmium-birth outcome associations among low and high adherers to the Mediterranean diet. We found no evidence for effect measure modification by Mediterranean diet adherence in the association between prenatal cadmium exposure and either gestational age or birth weight. Among women who reported high adherence to a Mediterranean diet pattern during pregnancy, the magnitude of the association between prenatal cadmium exposure and birth weight (*β* = −126.46; 95%CI: −453.14, 200.22; *p* = 0.44) was indistinguishable from the *β* = −210.38 observed among all participants. Similarly, the association between prenatal cadmium exposure and birth weight among women with low Mediterranean adherence was not statistically significant (*β* = −64.76; 95% CI: −359.90, 230.37; *p* = 0.66) (Table 3). However, these risk estimates lacked precision as confidence intervals were wide. We also observed no evidence for effect measure modification by maternal Mediterranean diet adherence on the association between prenatal cadmium exposure and preterm birth (high adherence: *β*: 0.01; 95% CI: 0, 1.59; *p* = 0.20; low adherence: *β*: 0.07; 95% CI: 0.0008, 1.31; *p* = 0.14) (Table 4). As expected from stratified analyses, including the interaction term of the Mediterranean diet adherence score and cadmium exposure did not alter these findings. The p-values for the interaction terms of cadmium and MDS in the overall birthweight or preterm birth models were not significant (*p* > 0.15). Defining “high Mediterranean adherence” with a more stringent cut-off of MDS of 5, 6 or 7 did alter these findings (data not shown).

### 3.3. Exploratory Analyses: Apgar Scores and PI

In our exploratory analyses we found no association between elevated prenatal cadmium exposure and Apgar scores (*β* = −0.009; 95% CI: −0.13, 0.12; *p* = 0.89) or PI (*β* = −0.03; 95% CI: −0.11, 0.06; *p* = 0.51). Results available in Appendix A.

## 4. Discussion

Understanding the effects of cadmium in early life is important, as this toxic metal is ubiquitous in the environment. With no set upper threshold for children, the accepted tolerable limits based on body weight are likely detrimental to children whose body weight is also smaller. In these analyses, we evaluated the extent to which adherence to a Mediterranean diet modified the association of cadmium and poor birth outcomes. We found that elevated prenatal cadmium exposure was associated with a lower birth weight compared to infants born to mothers with average or low cadmium exposure during pregnancy. These associations persisted after further adjusting for other co-occurring toxic metals that have been previously associated with these poor birth outcomes. Furthermore, after removing gestational age as a confounder, the association between prenatal cadmium exposure and birth weight remained significant. However, the association between prenatal cadmium exposure and preterm birth lost significance after removing birth weight. We found no association between prenatal cadmium exposure and Apgar score or infant PI at birth.

Our findings are consistent with previous data from our group and others that have demonstrated that prenatal exposure to cadmium, at non-occupational levels, is associated with lower birth weight [10,11,12] and is not associated with preterm birth [10,38]. These findings are, however, not consistent with the hypothesis that dietary patterns rich in iron, selenium, and folate may mitigate exposure. We did not find evidence to support our hypothesis that adherence to a Mediterranean diet prenatally modifies the associations between cadmium exposure and poor birth outcomes, regardless of the cut-off used to define “high Mediterranean adherence”. While sample size limits inference, these data suggest that, in this population, at these cadmium levels, adherence to a Mediterranean dietary pattern may not modify the effects of prenatal cadmium exposure on birth outcomes. We were also interested in understanding whether the association between prenatal cadmium exposure and adverse birth outcomes was related to growth restriction or shortened gestation. In our exploratory analysis, the association between cadmium exposure and birth weight remained significant after excluding gestational age from the model; however, the association between cadmium exposure and preterm birth was no longer significant after excluding birth weight. This suggests that cadmium exposure may influence birth weight through growth restriction rather than shortened gestation.

To our knowledge, this is the first attempt to determine the effects of the Mediterranean diet on the association between prenatal cadmium concentrations and documented poor birth outcomes in humans. Previous animal- and cell-based studies have shown that a dietary intake of iron, calcium, selenium, and folate can reduce toxicity from cadmium exposure [20,21]. The Mediterranean diet pattern has been found to be a rich source of iron, folate, and selenium [39]. Thus, our analysis showing that adherence to a Mediterranean diet pattern during pregnancy did not change the association between prenatal cadmium exposure and birth weight or preterm birth was surprising. The average intake of selenium in the US is high at 108.5 mcg/day [40], the intake of iron and calcium from food is lower than recommended at 11.5–13.7 mg/day [40] and 748 to 968 mg/day (females) [41], respectively, and the intake of folate is insufficient for pregnancy at 455 mcg DFE/day (females) [40]. It is possible that usual eating patterns in the US may not provide sufficient amounts of these nutrients to mitigate elevated cadmium exposure; therefore, diet interventions that encourage following a diet pattern with higher levels of iron, calcium, and folate may be warranted. Interventions focused on dietary modifications may hold better prospects for implementation and adherence in exposed populations when compared to interventions focused on costly landscape remediation. Additionally, dietary intervention does not carry the health risks associated with cadmium chelation using agents such as EDTA.

The inability to find associations could be due to one of several possibilities, some related to how cadmium is estimated, and others related to the measurement of diet. For example, maternal circulating levels of cadmium may not reflect cadmium levels that the offspring may be exposed to, as there is evidence in support of cadmium being sequestered by the placenta [42]. Secondly, because in the United States the main source of cadmium in the diet is lettuce, milk and cookies [43], it is possible that the additional exposure to cadmium may overwhelm the nutritive benefits of this diet. It is also possible that women’s intake in our sample may not represent a true Mediterranean diet pattern, even at high MDS values, as food choices and availability may differ by country [44]. Additionally, it may be that adherence to a Mediterranean diet pattern does reduce the effects of cadmium exposure on lower birth weight, yet we were underpowered to detect the associations. Although we were unable to establish a modifying effect of Mediterranean diet on the association between prenatal cadmium exposure and birth outcomes, the Mediterranean diet has been shown to protect against a number of diseases and inflammatory processes in the body [45], thus providing a rationale for its continued study. Given the ubiquity of this toxic metal in the environment, the effects of cadmium on birth outcomes, and the plausibility that a Mediterranean diet may mitigate the adverse effects, repeating these analyses in larger data sets is warranted.

Our study findings should be interpreted in the context of the study limitations. In addition to being underpowered to detect a significant effect measure modification that may have existed, both the measurement of Mediterranean diet adherence and our inability to “remove” the effects of dietary items such as lettuce and milk, which are major sources of cadmium in the US, is a limitation. An analysis of these relationships in different populations may clarify these findings. Furthermore, although implausible values were excluded from analyses, the Mediterranean diet was computed from self-reported diet data, which may have led to misclassified food intake that may be further biased by social desirability. This may have led to the under-reporting of unhealthy foods and the over-reporting of healthy foods. An additional limitation is the use of the MDS to assess diet. Although widely used and associated with a number of health outcomes, the MDS does not assess many foods that are thought to be “detrimental”, such as sugar or highly processed convenience foods.

Despite this, our study also exhibits strengths. We used prospectively collected data, therefore we can establish the timing of exposure, modifiers, and outcomes. Another important strength of this study is that it assessed maternal dietary patterns rather than the intake of single nutrients. Humans consume most of their nutrients through foods and combinations of nutrients, therefore it is important to assess the role of dietary patterns in the potential mitigation of negative consequences from toxic exposures.

## 5. Conclusions

These limitations notwithstanding, this study contributes to the growing literature on the effects of toxic exposures during pregnancy and adds information on the potential role of diet in preventing adverse birth outcomes. Although our study did not support that maternal adherence to a Mediterranean diet pattern may mitigate exposure, it is possible that other dietary patterns may in fact help mitigate the association between elevated prenatal cadmium exposure and lower birth weights. Future research should focus on finding dietary patterns that can mitigate prenatal cadmium risk and that can be easily translatable into public health recommendations.

## Figures and Tables

**Table 1 toxics-08-00090-t001:** Sociodemographic characteristics of study participants by birth weight.

Characteristics	Overall	Low Birth Weight (<2500 g)	Non-Low Birth Weight (≥2500 g)
Ethnicity	N (%)	N (%)	N (%)
White	84 (28.2)	3 (17.6)	81 (28.8)
Black	108 (36.2)	11 (64.7)	97 (34.5)
Hispanic	94 (31.5)	2 (11.8)	92 (32.7)
Other	12 (4)	1 (5.9)	11 (3.9)
Maternal obesity before pregnancy			
<30	219 (73.5)	13 (76.4)	206 (73.3)
30+	76 (25.5)	4 (23.5)	72 (25.6)
Missing	3 (1.0)	--	3 (1.1)
Sex			
Male	149 (50.0)	8 (47.0)	141 (50.2)
Female	149 (50.0)	9 (52.9)	140 (49.8)
Gestational age at delivery			
<37 weeks	18 (6.0)	12 (70.6)	6 (2.1)
37+ weeks	280 (94.0)	5 (29.4)	275 (97.8)
Cigarette smoking during pregnancy			
No	242 (81.2)	9 (52.9)	233 (82.9)
Yes	46 (15.4)	8 (47.1)	38 (13.5)
Missing	10 (3.4)	--	10 (3.6)
Maternal educational attainment			
College Graduate and some college	134 (45.0)	6 (3.5)	128 (45.6)
High school or less	164 (55.0)	11 (64.7)	153 (54.4)
Cadmium categories			
Low	224 (75.2)	8 (47.1)	216 (76.9)
High	74 (24.8)	9 (52.9)	65 (23.1)
Cadmium concentrations (Median and interquartile range)			
Cd	0.24 (0.34)	0.67 (0.61)	0.23 (0.29)
Mediterranean diet adherence			
Low (≤4)	92 (30.9)	8 (47.1)	84 (29.9)
High (>4)	92 (30.9)	3 (17.6)	89 (31.7)
Missing	114 (38.2)	6 (35.3)	108 (38.4)

-- No missing data in this category.

**Table 2 toxics-08-00090-t002:** Sociodemographic characteristics of study participants by Mediterranean diet adherence.

Characteristics	Overall	Low Mediterranean Adherence (≤4)	High Mediterranean Adherence (>4)
Ethnicity	N (%)	N (%)	N (%)
White	70 (37.8)	24 (26.1)	46 (49.5)
Black	54 (29.2)	38 (41.3)	16 (17.2)
Hispanic	51 (27.6)	26 (28.2)	25 (26.9)
Other	10 (5.4)	4 (4.3)	6 (6.4)
Maternal obesity before pregnancy			
<30	143 (77.3)	69 (75.0)	74 (79.6)
30+	41 (22.2)	22 (23.9)	19 (20.4)
Missing	1 (0.5)	1 (1.1)	--
Sex			
Male	101 (54.6)	52 (56.5)	49 (52.7)
Female	84 (45.4)	40 (43.5)	44 (47.3)
Gestational age at delivery			
<37 weeks	13 (7.0)	9 (9.8)	4 (4.3)
37+ weeks	172 (93.0)	83 (90.2)	89 (95.7)
Cigarette smoking during pregnancy			
No	158 (85.4)	73 (79.3)	85 (91.4)
Yes	22 (11.9)	16 (17.4)	6 (6.5)
Missing	5 (2.7)	3 (3.3)	2 (2.1)
Maternal educational attainment			
College Graduate and some college	86 (46.5)	53 (57.6)	33 (35.5)
High school or less	99 (53.5)	39 (41.9)	60 (64.5)
Birth Weight			
<2500 g	11 (6.0)	8 (8.7)	3 (3.2)
2500+ grams	173 (93.5)	84 (91.3)	89 (95.7)
Missing	1 (0.5)	--	1 (1.1)
Cadmium categories			
Low	147 (79.5)	69 (75.0)	78 (83.9)
High	38 (20.5)	23 (25.0)	15 (16.1)
Metal concentrations (Median and interquartile range)			
Cd	0.1892 (0.34)	0.2319 (0.36)	0.1723 (0.21)

-- No missing data in this category.

**Table 3 toxics-08-00090-t003:** Regression coefficients and 95% confidence intervals for the association/relationship between, cadmium exposure and weight.

Factor	β	95% CI	*p*
Birth weight ^a^	−210.38	(−332.00, −88.78)	0.0008
Birth weight, no gestational age ^d^	−161.596	(−311.33, −11.86)	0.04
Birth weight, low Med adherence (≤4) ^b^	−64.76	(−359.89, 230.37)	0.66
Birth Weight, high Med adherence (>4) ^c^	−126.46	(−453.14, 200.22)	0.44

^a^ Adjusted for smoking during pregnancy, prepregnancy BMI, gestational age and sex of the infant. ^b^ Adjusted for smoking during pregnancy, prepregnancy BMI, gestational age and sex of the infant, among mothers with a Mediterranean diet score at or below 4. ^c^ Adjusted for smoking during pregnancy, prepregnancy BMI, gestational age and sex of the infant, among mothers with a Mediterranean diet score above 4. ^d^ Adjusted for smoking during pregnancy, prepregnancy BMI, and sex of the infant.

**Table 4 toxics-08-00090-t004:** Odds ratio and 95% confidence intervals of the association between elevated prenatal cadmium exposure and preterm birth.

Factor	Odds Ratio	95% CI	*p*
Preterm birth ^a^	0.11	(0.01, 0.72)	0.04
Preterm birth, no birth weight ^d^	1.20	(0.37, 3.31)	0.74
Preterm birth, low Med adherence (≤4) ^b^	0.07	(0.0008, 1.31)	0.14
Preterm birth, high Med adherence (>4) ^c^	0.01	(0, 1.59)	0.20

^a^ Adjusted for smoking during pregnancy, prepregnancy BMI, birth weight and sex of the infant. ^b^ Adjusted for smoking during pregnancy, prepregnancy BMI, birth weight and sex of the infant, among mothers with a Mediterranean diet score at or below 4. ^c^ Adjusted for smoking during pregnancy, prepregnancy BMI, birth weight and sex of the infant, among mothers with a Mediterranean diet score above 4. ^d^ Adjusted for smoking during pregnancy, prepregnancy BMI, and sex of the infant.

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
