# Peer review of "Associations between Maternal Cadmium Exposure with Risk of Preterm Birth and Low Birth Weight: Effect of Mediterranean Diet Adherence on Affected Prenatal Outcomes"

_toxics, 2020, doi:10.3390/toxics8040090_

Round 1
Reviewer 1 Report
In this study, Gonzalez-Nahm et all. aim to evaluate if higher adherence to a Mediterranean diet during pregnancy may mitigate the adverse effect of maternal cadmium exposure in terms of birth weight and gestational age.
The paper is of interest but it needs to be improved to be published.
- Please improve the title including all the outcomes explored.
- Material and method section needs to be better described and improved furnishing more methodological details and references. In particular, the Authors should furnish more details related to sample size calculation.
- The table 1 organization is not clear. At the beginning, the data are related to Mediterranean diet adherence and then data are related to birth weight. Is this right? If yes please produce two tables with more specific titles.
- Why in the first part of the table 1 are reported 185 newborns instead of 184?
- Always in table 1, percentages are lacking for the second and third column.
- Besides evaluating the birth weight, it would be interesting to have some more indices evaluated such as the APGAR score birth length, head circumference, and the ponderal index.
- What is the relation, if any, between maternal cadmium blood concentration and the maternal weight?
- Have the authors evaluated if maternal blood cadmium correlates with cadmium in cordonal blood?
- Please show all quantiles of the cadmium distribution
- The author should better explain how variation in Mediterranean Diet score associate to prenatal cadmium exposure.
- Please format all references following the MDPI instructions. At lane 36 and 71, for example, numbers are out of brackets. Please check all reference appropriateness.
- Please check the appropriateness of reference 35 (lane 270).
- Data relating Mediterranean diet adherence and preterm birth should be explored and discussed in the introduction section and in discussion section (see Peraita-Costa I, Llopis-González A, Perales-Marín A, Diago V, Soriano JM, Llopis-Morales A, Morales-Suárez-Varela M. Maternal profile according to Mediterranean diet adherence and small for gestational age and preterm newborn outcomes. Public Health Nutr. 2020 Apr 29:1-13. doi: 0.1017/S1368980019004993. Epub ahead of print. PMID: 32345384)
- Please update and discuss more recent literature data. The following important references are lacking.
- (1)Is the Concentration of Cadmium, Lead, Mercury, and Selenium Related to Preterm Birth?
- Yıldırım E, Derici MK, Demir E, Apaydın H, Koçak Ö, Kan Ö, Görkem Ü. Biol Trace Elem Res. 2019 Oct;191(2):306-312. doi: 10.1007/s12011-018-1625-2. Epub 2019 Jan 2. PMID: 30600504
- (2)Association of maternal serum cadmium level during pregnancy with risk of preterm birth in a Chinese population.
Wang H, Liu L, Hu YF, Hao JH, Chen YH, Su PY, Yu Z, Fu L, Tao FB, Xu DX. Environ Pollut. 2016 Sep;216:851-857. doi: 10.1016/j.envpol.2016.06.058. Epub 2016 Jul 2. PMID: 27381872
- (3)Maternal cadmium levels during pregnancy associated with lower birth weight in infants in a North Carolina cohort.
Johnston JE, Valentiner E, Maxson P, Miranda ML, Fry RC. PLoS One. 2014 Oct 6;9(10):e109661. doi: 10.1371/journal.pone.0109661. eCollection 2014. PMID: 25285731
Reviewer 2 Report
Association between maternal cadmium and birth weight and the Mediterranean diet
Gonzalez-Nahm et al aimed to explore the potential mitigating role of adherence to a Mediterranean diet pattern during pregnancy in the association between elevated prenatal cadmium exposure and birth outcomes.
The paper has a potential interest but must be substantially improved to reach a publishable form.
- P2, L70-82. This paragraph must be re-written to give more information about the selection process. There are some sentences that are not understandable.
- Please format all references in the text in the same way.
- P3, L123-16. The notation should be uniform and unequivocally in the text and tables.
- P3, L131. …in the top 25th” Is this right?
- Authors mention that confounder have been chosen following the BIC criteria. But the BIC criteria is a statistical diagnosis not a biologic consideration. You must do an accurate selection of variables satisfying the three characteristics of a confounder on biological basis.
- Tables are presented in a singular way. Authors state that aimed to explore the association between elevated prenatal cadmium exposure with birth outcomes, but the data are presented by scores of adherence to Mediterranean Diet. Please draw a table with rows displaying the variables and columns displaying the outcomes. One table for each outcome.
- Please show all quantiles of the cadmium distribution
- In the statistical analysis section, the first paragraph says: “We used linear regression to assess the association between elevated cadmium exposure during pregnancy and birth weight and logistic regression to assess the association between elevated cadmium exposure and preterm birth.” This implies that performing linear regression we obtain coefficients in the natural scale variables and outcome have been measured. On the other way, by performing logistic regression we obtain coefficients measured in the logarithmic scale of variables and their effect on the Odds Ratio. These two statistical procedures that have a so different interpretation have been combined in the same table under the headings β, 95% CI and p or under the headings Coefficient, 95% CI and p. As they are different measures they should be showed in different tables and discussed separately.
- Assess if you have estimated effect modification of Mediterranean Diet on prenatal cadmium exposure.
- In table 1, N and % constitute the columns subheadings but percentages are missing for the second and third column.
- Although in the objective authors state the objective of exploring the mitigating role of Mediterranean Diet there is no analysis about this objective. May be authors wanted to explore the effect of prenatal cadmium exposure on birth outcome stratifying by adherence to Mediterranean Diet?
- Please include an analysis with the exposure variable in continuous form.
- Please include in the title the two outcomes you are exploring.
Reviewer 3 Report
Overall a well done study with soundful outcome
The text (introduction) contains numbers without context, please
change thad
Author Response
Reviewer 3
Overall a well done study with soundful outcome
The text (introduction) contains numbers without context, please change that
Thank you. We have corrected this issue and have ensured that numbers indicating references are contained in brackets.
Reviewer 4 Report
This MS has serious gaps, scientific errors, very weak language and inappropriate experimental design. The title has no idea what authors want to convey. The summary is imperceptible. The terms "diet", nutrients, nutritive metals should be replaced by an appropriate language. There is evidence to support the fact that the meditteranic diet is suitable for maternal health, pregnancy and progeny. The MS lacks references. In the discussion there is also a lack of comparison with data from other authors. An ethical statement is required for human studies.
Other comments:
Please include references on mediterranean diet to Support its value for pregnacy and normal outcomes.
L.15 – Please rephrase the sentence, since Cd is toxic: “Although …minerals can be toxic”.
L.44 – why authors associate obesity and zebrafish?
L.46-51 – These aspects related to low birth weight are associated with Cd?
L.55 – Which is the meaning of nutritive metals?
L.76 – How cadmium was measured?
L.127 – Please reduce statistical section.
L.242 - Which is the meaning of : “To our knowledge, this is the first attempt to determine the effects of the Mediterranean diet—a diet rich in +2 ions?
Round 2
Reviewer 1 Report
T
The manuscript has been sufficiently improved.
However, the title is not clear yet. It should be further improved.
I suggest: "Association between maternal cadmium exposure with the risk of preterm birth and low birth weight. Effect of Mediterranean diet adherence on affected prenatal outcomes"
English needs to be revisited to make sentences more clear and easy-to-read.
Author Response
Thank you for your review. We have changed the title per the reviewer'suggestion to: "Association between maternal cadmium exposure with the risk of preterm birth and low birth weight. Effect of Mediterranean diet adherence on affected prenatal outcomes"
We have conducted extensive editing to the text for improved readability. Edits are in track changes.
Reviewer 2 Report
Authors have addressed all issues raised by me
Author Response
Thank you. We have edited the manuscript for improved readability. All edits have been made using track changes.
Reviewer 4 Report
Authors have done most all changes to improve the MS. Minor changes in english language are required.
Author Response
Thank you for your review. We have extensively edited the manuscript to ensure the results are more clearly presented and for improved readability. All edits have been made using track changes.